# Novelty Search in Representational Space for Sample Efficient Exploration

**Ruo Yu Tao[1, 2, *], Vincent François-Lavet[1, 2], Joelle Pineau[1, 2]**
[1] McGill University
[2] Mila, Quebec Artificial Intelligence Institute
[*] ruo.tao@mail.mcgill.ca

## Abstract

We present a new approach for efficient exploration which leverages a low-dimensional encoding of the environment learned with a combination of model-based and model-free objectives. Our approach uses intrinsic rewards that are based on the distance of nearest neighbors in the low dimensional representational space to gauge novelty. We then leverage these intrinsic rewards for sample-efficient exploration with planning routines in representational space for hard exploration tasks with sparse rewards. One key element of our approach is the use of information theoretic principles to shape our representations in a way so that our novelty reward goes beyond pixel similarity. We test our approach on a number of maze tasks, as well as a control problem and show that our exploration approach is more sample-efficient compared to strong baselines.

## 1 Introduction

In order to solve a task efficiently in Reinforcement Learning (RL), one of the main challenges is to gather informative experiences via an efficient exploration of the state space. A common approach to exploration is to leverage intrinsic rewards correlated with some metric or score for novelty (Schmidhuber, 2010; Stadie et al., 2015; Houthooft et al., 2016). With intrinsic rewards, an agent can be incentivized to efficiently explore its state space. A direct approach to calculating these novelty scores is to derive a reward based on the observations, such as a count-based reward (Bellemare et al., 2016; Ostrovski et al., 2017) or a prediction-error based reward (Burda et al., 2018b). However, an issue occurs when measuring novelty directly from the raw observations, as some information in pixel space (such as randomness or backgrounds) may be irrelevant. In this case, if an agent wants to efficiently explore its state space it should only focus on meaningful and novel information.

In this work, we propose a method of sample-efficient exploration by leveraging intrinsic rewards in a meaningful latent state space. To build a meaningful state abstraction, we view Model-based RL (MBRL) from an information theoretic perspective - we optimize our dynamics learning through the Information Bottleneck (Tishby et al., 2000) principle. We also combine both model-based and model-free components through a joint representation. This method encodes high-dimensional observations into lower-dimensional representations such that states that are close in *dynamics* are brought close together in representation space (François-Lavet et al., 2018). We also add additional constraints to ensure that a measure of distance between abstract states is meaningful. We leverage these properties of our representation to formulate a novelty score based on Euclidean distance in low-dimensional representation space and we then use this score to generate intrinsic rewards that we can exploit for efficient exploration.

One important element of our exploration algorithm is that we take a Model Predictive Control (MPC) approach (Garcia et al., 1989) and perform actions only after our model is sufficiently accurate (and hence ensure an accurate novelty heuristic). Through this training scheme, our agent is

also able to learn a meaningful representation of its state space in a sample-efficient manner. The code with all experiments is available [1].

## 2  Problem setting

An agent interacts with its environment over discrete timesteps, modeled as a Markov Decision Process (MDP), defined by the 6-tuple $(\mathcal{S}, \mathcal{S}_0, \mathcal{A}, \tau, \mathcal{R}, \mathcal{G})$ (Puterman, 1994). In this setting, $\mathcal{S}$ is the state space, $\mathcal{S}_0$ is the initial state distribution, $\mathcal{A}$ is the discrete action space, $\tau : \mathcal{S} \times \mathcal{A} \to \mathcal{S}$ is the transition function that is assumed deterministic (with the possibility of extension to stochastic environments with generative methods), $R : \mathcal{S} \times \mathcal{A} \to \mathcal{R}$ is the reward function ($\mathcal{R} = [-1, 1]$), $\mathcal{G} : \mathcal{S} \times \mathcal{A} \to [0, 1)$ is the per timestep discount factor. At timestep $t$ in state $s_t \in \mathcal{S}$, the agent chooses an action $a_t \in \mathcal{A}$ based on policy $\pi : \mathcal{S} \times \mathcal{A} \to [0, 1]$, such that $a_t \sim \pi(s_t, \cdot)$. After taking $a_t$, the agent is in state $s_{t+1} = \tau(s_t, a_t)$ and receives reward $r_t \sim R(s_t, a_t)$ and a discount factor $\gamma_t \sim \mathcal{G}(s_t, a_t)$. Over $n$ environment steps, we define the buffer of previously visited states as $\mathcal{B} = (s_1, \ldots, s_n)$, where $s_i \in \mathcal{S} \ \forall i \in \mathbb{N}$. In RL, the usual objective is to maximize the sum of expected future rewards $V_\pi(s) = \mathbb{E}_\pi \left[ r_t + \sum_{i=1}^{\infty} \left( \prod_{j=0}^{i-1} \gamma_{t+j} \right) r_{t+i} | s = s_t \right]$.

To learn a policy $\pi$ that maximizes the expected return, an RL agent has to efficiently explore its environment (reach novel states in as few steps as possible). In this paper, we consider tasks with sparse rewards or even no rewards, and are interested in exploration strategies that require as few steps as possible to explore the state space.

## 3  Abstract state representations

We focus on learning a lower-dimensional representation of state when our state (or observations in the partially observable case (Kaelbling et al., 1998)) is high-dimensional (Dayan, 1993; Tamar et al., 2016; Silver et al., 2016; Oh et al., 2017; de Bruin et al., 2018; Ha and Schmidhuber, 2018; François-Lavet et al., 2018; Hafner et al., 2018; Gelada et al., 2019).

### 3.1  Information Bottleneck

We first motivate our methods for model learning. To do so, we consider the *Information Bottleneck* (IB) (Tishby et al., 2000) principle. Let $Z$ denote the original source message space and $\tilde{Z}$ denote its compressed representation. As opposed to traditional lossless compression where we seek to find corresponding encodings $\tilde{Z}$ that compresses all aspects of $Z$, in IB we seek to preserve only *relevant* information in $\tilde{Z}$ with regards to another relevance variable, $Y$. For example when looking to compress speech waveforms ($Z$) if our task at hand is speech recognition, then our relevance variable $Y$ would be a transcript of the speech. Our representation $\tilde{Z}$ would only need to maximize relevant information about the transcript $Y$ instead of its full form including tone, pitch, background noise etc. We can formulate this objective by minimizing the following functional with respect to $p(\tilde{z} \mid z)$:

$$\mathcal{L}(p(\tilde{z} \mid z)) = I[Z; \tilde{Z}] - \beta I[\tilde{Z}; Y]$$

where $I[\cdot; \cdot]$ is the *Mutual Information* (MI) between two random variables. $\beta$ is the Lagrange multiplier for the amount of information our encoding $\tilde{Z}$ is allowed to quantify about $Y$. This corresponds to a trade-off between minimizing the encoding rate $I[Z; \tilde{Z}]$ and maximizing the mutual information between the encoding and our random variable $Y$.

We now apply this principle to representation learning of state in MBRL. If our source message space is our state $S'$ and our encoded message is $X'$, then to distill the most relevant information with regards to the dynamics of our environment one choice of relevance variable is $\{X, A\}$, i.e. our encoded state in the previous timestep together with the presence of an action. This gives us the functional

$$\mathcal{L}(p(x' \mid s')) = I[S'; X'] - \beta I[X'; \{X, A\}]. \tag{1}$$

In our work, we look to find methods to minimize this functional for an encoding that maximizes the predictive ability of our dynamics model.

We first aim to minimize our encoding rate $I[S'; X']$. Since encoding rate is a measure of the amount of bits transmitted per message $S'$, representation dimension is analogous to number of bits per message. This principle of minimizing encoding rate guides our selection of representation dimension

- for every environment, we try to choose the smallest representation dimension possible such that the representation can still encapsulate model dynamics as we understand them. For example, in a simple Gridworld example, we look to only encode agent position in the grid-world.

Now let us consider the second term in Equation 1. Our goal is to learn an optimally predictive model of our environment. To do so we first consider the MI between the random variable denoting our state representation $X$, in the presence of the random variable representing actions $A$ and the random variable denoting the state representation in the next timestep $X'$ (Still, 2009). Note that MI is a metric and is symmetric:

$$I[\{X, A\} \; ; \; X'] = \mathbb{E}_{p(x',x,a)} \left[ \log \left( \frac{p(x' \mid x, a)}{p(x')} \right) \right] = H[X'] - H[X' \mid X, A] \qquad (2)$$

This quantity is a measure of our dynamics model's predictive ability. If we consider the two entropy terms (denoted $H[\cdot]$), we see that $H[X']$ constitutes the entropy of our state representation and $H[X' \mid X, A]$ as the entropy of the next state $X'$ given our current state $X$ and an action $A$. Recall that we are trying to minimize $I[X'; S']$ and maximize $I[X'; \{X, A\}]$ with respect to some encoding function $X = e(S)$. In the next section, we describe our approach for this encoding function as well as dynamics learning in MBRL.

## 3.2 Encoding and dynamics learning

For our purposes, we use a neural encoder $\hat{e} : \mathcal{S} \rightarrow \mathcal{X}$ parameterized by $\theta_{\hat{e}}$ to map our high-dimensional state space into lower-dimensional abstract representations, where $\mathcal{X} \subseteq \mathbb{R}^{n_{\mathcal{X}}}$. The dynamics are learned via the following functions: a transition function $\hat{\tau} : \mathcal{X} \times A \rightarrow \mathcal{X}$ parameterized by $\theta_{\hat{\tau}}$, a reward function $\hat{r} : \mathcal{X} \times A \rightarrow [-1, 1]$ parameterized by $\theta_{\hat{r}}$, and a per timestep discount factor function $\hat{\gamma} : \mathcal{X} \times A \rightarrow [0, 1)$ parameterized by $\theta_{\hat{\gamma}}$. This discount factor is only learned to predict terminal states, where $\gamma = 0$.

In order to leverage all past experiences, we use an off-policy learning algorithm that samples transition tuples $(s, a, r, \gamma, s')$ from a replay buffer. We first encode our current and next states with our encoder to get $x \leftarrow \hat{e}(s; \theta_{\hat{e}})$, $x' \leftarrow \hat{e}(s'; \theta_{\hat{e}})$. The Q-function is learned using the DDQN algorithm (van Hasselt et al., 2015), which uses the target:

$$Y = r + \gamma Q(\hat{e}(s'; \theta_{\hat{e}-}), \operatorname*{argmax}_{a' \in \mathcal{A}} Q(x', a'; \theta_Q); \theta_{Q-}),$$

where $\theta_{Q-}$ and $\theta_{\hat{e}-}$ are parameters of an earlier buffered Q-function (or our target Q-function) and encoder respectively. The agent then minimizes the following loss:

$$L_Q(\theta_Q) = (Q(x, a; \theta_Q) - Y)^2.$$

We learn the dynamics of our environment through the following losses:

$$L_R(\theta_{\hat{e}}, \theta_{\hat{r}}) = |r - \hat{r}(x, a; \theta_{\hat{r}})|^2 , \; L_{\mathcal{G}}(\theta_{\hat{e}}, \theta_{\hat{\gamma}}) = |\gamma - \hat{\gamma}(x, a; \theta_{\hat{\gamma}})|^2$$

and our transition loss

$$L_\tau(\theta_{\hat{e}}, \theta_{\hat{\tau}}) = ||[x + \hat{\tau}(x, a; \theta_{\hat{\tau}})] - x'||_2^2. \qquad (3)$$

Note that our transition function learns the difference (given an action) between previous state $x$ and current state $x'$. By jointly learning the weights of the encoder and the different components, the abstract representation is shaped in a meaningful way according to the dynamics of the environment. In particular, by minimizing the loss given in Equation 3 with respect to the encoder parameters $\theta_{\hat{e}}$ (or $p(x \mid s)$), we minimize our entropy $H[X'|X, A]$.

In order to maximize the entropy of our learnt abstracted state representations $H[X']$, we minimize the expected pairwise Gaussian potential (Borodachov et al., 2019) between states:

$$L_{d1}(\theta_{\hat{e}}) = \mathbb{E}_{s_1, s_2 \sim p(s)} \left[ exp(-C_{d1}||\hat{e}(s_1; \theta_{\hat{e}}) - \hat{e}(s_2; \theta_{\hat{e}})||_2^2) \right] \qquad (4)$$

with $C_{d1}$ as a hyperparameter. Losses in Equation 3 and Equation 4 are reminiscent of the model-based losses in François-Lavet et al. (2018) and correspond respectively to the *alignment* and *uniformity* contrastive loss formulation in Wang and Isola (2020), where alignment ensures that similar states are close together (in encoded representation space) and uniformity ensures that all states are spread uniformly throughout this low-dimensional representation space.

The losses $L_\tau(\theta_{\hat{e}})$ and $L_{d1}(\theta_{\hat{e}})$ maximizes the $I[\{X, A\}; X']$ term and selecting smaller dimension for our representation minimizes $I[X', S']$. Put together, our method is trying to minimize $\mathcal{L}(p(x'|s'))$ as per Equation 1.

### 3.3 Distance measures in representational space

For practical purposes, since we are looking to use a distance metric within $\mathcal{X}$ to leverage as a score for novelty, we ensure well-defined distances between states by constraining the $\ell_2$ distance between two consecutive states:

$$L_{csc}(\theta_{\hat{e}}) = max(\|\hat{e}(s_1; \theta_e) - \hat{e}(s_2; \theta_e)\|_2 - \omega, 0) \tag{5}$$

where $L_{csc}$ is a soft constraint between consecutive states $s_1$ and $s_2$ that tends to enforce two consecutive encoded representations to be at a distance $\omega$ apart. We add $L_{csc}$ to ensure a well-defined $\ell_2$ distance between abstract states for use in our intrinsic reward calculation (a discussion of this loss is provided in Appendix B). We discuss how we use $\omega$ to evaluate model accuracy for our MPC updates in Appendix A. Finally, we minimize the sum of all the aforementioned losses through gradient descent:

$$\mathcal{L} = L_R(\theta_{\hat{e}}, \theta_{\hat{r}}) + L_{\mathcal{G}}(\theta_{\hat{e}}, \theta_{\hat{\gamma}}) + L_{\tau}(\theta_{\hat{e}}, \theta_{\hat{\tau}}) + L_Q(\theta_Q) + L_{d1}(\theta_{\hat{e}}) + L_{csc}(\theta_{\hat{e}}). \tag{6}$$

Through these losses, the agent learns a low-dimensional representation of the environment that is meaningful in terms of the $\ell_2$ norm in representation space. We then employ a planning technique that combines the knowledge of the model and the value function which we use to maximize intrinsic rewards, as detailed in the next section and Section 4.3.

## 4 Novelty Search in abstract representational space

Our approach for exploration uses *intrinsic motivation* (Schmidhuber, 1990; Chentanez et al., 2005; Achiam and Sastry, 2017) where an agent rewards itself based on the fact that it gathers interesting experiences. In a large state space setting, states are rarely visited and the count for any state after $n$ steps is almost always 0. While Bellemare et al. (2016) solves this issue with density estimation using pseudo-counts directly from the high-dimensional observations, we aim to estimate some function of novelty in our learnt lower-dimensional representation space.

### 4.1 Sparsity in representation space as a measure for novelty

Through the minimization of Equation 1, states that are close together in dynamics are pushed close together in our abstract state space $\mathcal{X}$. Ideally, we want an agent that efficiently explores the *dynamics* of its environment. To do so, we reward our agent for exploring areas in lower-dimensional representation space that are less visited and ideally as far apart from the dynamics that we currently know.

Given a point $x$ in representation space, we define a reward function that considers the *sparsity* of states around $x$ - we do so with the average distance between $x$ and its $k$-nearest-neighbors in its visitation history buffer $\mathcal{B}$:

$$\hat{\rho}_{\mathcal{X}}(x) = \frac{1}{k}\sum_{i=1}^{k} d(x, x_i), \tag{7}$$

where $x \doteq \hat{e}(s; \theta_{\hat{e}})$ is a given encoded state, $k \in \mathbb{Z}^+$, $d(\cdot, \cdot)$ is some distance metric in $\mathbb{R}^{n_{\mathcal{X}}}$ and $x_i \doteq \hat{e}(s_i; \theta_{\hat{e}})$, where $s_i \in \mathcal{B}$ for $i = 1 \dots k$ are the $k$ nearest neighbors (by encoding states in $\mathcal{B}$ to representational space) of $x$ according to the distance metric $d(\cdot, \cdot)$. Implicit in this measure is the reliance on the agent's visitation history buffer $\mathcal{B}$.

An important factor in this score is which distance metric to use. With the losses used in Section 3, we use $\ell_2$ distance because of the structure imposed on the abstract state space with Equations 4 and 5.

As we show in Appendix D, this novelty reward is reminiscent of *recoding probabilities* (Bellemare et al., 2016; Cover and Thomas, 2012) and is in fact inversely proportional to these probabilities, suggesting that our novelty heuristic estimates visitation count. This is also the same score used to gauge "sparseness" in behavior space in Lehman and Stanley (2011).

With this reward function, we present the pseudo-code for our exploration algorithm in Algorithm 1.

**Algorithm 1:** The Novelty Search algorithm in abstract representational space.

---

**1 Initialization:** transition buffer $\mathcal{B}$, agent policy $\pi$;
**2** Sample $n_{init}$ initial random transitions, let $t = n_{init}$;
**3 while** $t \leq n_{max}$ **do**
      `// We update our dynamics model and Q-function every` $n_{freq}$ `steps`
**4**    **if** $t \mod n_{freq} == 0$ **then**
**5**        **while** $j \leq n_{iters}$ *or* $L_\tau \leq \left(\frac{\omega}{\delta}\right)^2$ **do**
**6**            Sample batch of transitions $(s, a, r_{extr}, r_{intr}, \gamma, s') \in \mathcal{B}$;
**7**            Train dynamics model with $(s, a, r_{extr}, \gamma, s')$;
**8**            Train Q-function with $(s, a, r_{extr} + r_{intr}, \gamma, s')$;
**9**        **end**
**10**        $\forall (s, a, r_{extr}, r_{intr}, \gamma, s') \in \mathcal{B}$, set $r_{intr} \leftarrow \hat{\rho}_{\mathcal{X}}(\hat{e}(s'; \theta_{\hat{e}}))$;
**11**    **end**
**12**    $a_t \sim \pi(s_t)$;
**13**    Take action in environment: $s_{t+1} \leftarrow \tau(s_t, a_t)$, $r_{t,extr} \leftarrow R(s_t, a_t)$, $\gamma_t \leftarrow \mathcal{G}(s_t, a_t)$;
**14**    Calculate intrinsic reward: $r_{t,intr} \leftarrow \hat{\rho}_{\mathcal{X}}(\hat{e}(s_{t+1}; \theta_{\hat{e}}))$
**15**    $\mathcal{B} \leftarrow \mathcal{B} \cup \{(s_t, a_t, r_{t,extr}, r_{t,intr}, \gamma_t, s_{t+1})\}$;
**16 end**

---

### 4.2 Asymptotic behavior

This reward function also exhibits favorable asymptotic behavior, as it decreases to 0 as most of the state space is visited. We show this in Theorem 1.

**Theorem 1.** *Assume we have a finite state space* $S \subseteq \mathbb{R}^d$, *history of states* $\mathcal{B} = (s_1, \ldots, s_N)$, *encoded state space* $\mathcal{X} \subseteq \mathbb{R}^{n_{\mathcal{X}}}$, *deterministic mapping* $f : \mathbb{R}^d \to \mathbb{R}^{n_{\mathcal{X}}}$ *and a novelty reward defined as* $\hat{\rho}_{\mathcal{X}}(x)$. *With an optimal policy with respect to the rewards of the novelty heuristic, our agent will tend towards states with higher intrinsic rewards. If we assume a communicating MDP setting (Puterman, 1994), we have that*

$$\lim_{N \to \infty} \hat{\rho}_{\mathcal{X}}(f(s)) = 0, \ \forall s \in S.$$

*Proof.* We prove this theorem in Appendix E. $\square$

### 4.3 Combining model-free and model-based components for exploration policies

Similarly to previous works (e.g. Oh et al., 2017; Chebotar et al., 2017), we use a combination of model-based planning with model-free Q-learning to obtain a good policy. We calculate rollout estimates of next states based on our transition model $\hat{\tau}$ and sum up the corresponding rewards, which we denote as $r : \mathcal{X} \times A \to [0, R_{max}]$ and can be a combination of both intrinsic and extrinsic rewards. We calculate expected returns based on the discounted rewards of our $d$-depth rollouts:

$$\hat{Q}^d(x, a) = \begin{cases} r(x, a) + \hat{\gamma}(x, a; \theta_{\hat{\gamma}}) \times \\ \max_{a' \in \mathcal{A}} \hat{Q}^{d-1}(\tau(x, a; \theta_{\hat{\tau}}), a'), & \text{if } d > 0 \\ Q(x, a; \theta_Q), & \text{if } d = 0 \end{cases} \tag{8}$$

Note that we simulate only $b$-best options at each expansion step based on $Q(x, a; \theta_Q)$, where $b \leq |\mathcal{A}|$. In this work, we only use full expansions. The estimated optimal action is given by

$$a^* = \operatorname*{argmax}_{a \in \mathcal{A}} \hat{Q}^d(x, a).$$

The actual action chosen at each step follows an $\epsilon$-greedy strategy ($\epsilon \in [0, 1]$), where the agent follows the estimated optimal action with probability $1 - \epsilon$ and a random action with probability $\epsilon$.

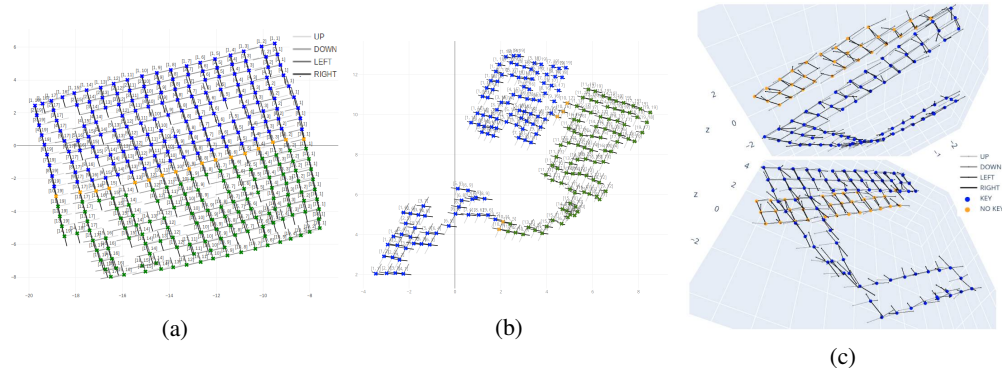

(a)                                        (b)

(c)

Figure 1: (a), (b): Plotting the full history of learned abstract representations of both open and 4-room labyrinth environments from Figures 7a and 7b after 500 environment steps. Colors denote which side of the maze the agent was in, grid coordinates and transitions are shown. (c): Two views of the same full history of learned abstract 3-dimensional representation of our multi-step maze after 300 steps. Orange and blue points denote states without and with keys respectively. Our agent is able to disentangle states where the agent has a key and when it doesn't as seen in the distance between the two groups of states. Meaningful information about the agent position is also maintained in the relative positions of states in abstract state space.

## 5 Experiments

We conduct experiments on environments of varying difficulty. All experiments use a training scheme where we first train parameters to converge on an accurate representation of the already experienced transitions before taking an environment step. We optimize the losses (over multiple training iterations) given in Section 3. We discuss all environment-specific hyperparameters in Appendix J.

### 5.1 Labyrinth exploration

We consider two $21 \times 21$ versions of the grid-world environment (Figure 7 in Appendix). The first is an open labyrinth grid-world, with no walls except for bordering walls. The second is a similar sized grid-world split into four connected rooms. In these environments the action space $\mathcal{A}$ is the set of four cardinal directions. These environments have no rewards or terminal states and the goal is to explore, agnostic of any task. We use two metrics to gauge exploration for this environment: the first is the ratio of states visited only once, the second is the proportion of total states visited.

#### 5.1.1 Open labyrinth

In the open labyrinth experiments (Figure 2a), we compare a number of variations of our approach with a random baseline and a count-based baseline (Bellemare et al., 2016) (as we can count states in this tabular setting). Variations of the policy include an argmax over state values ($d = 0$) and planning depths of $d \in \{1, 5\}$. All variations of our method outperform the two baselines in this task, with a slight increase in performance as planning depth $d$ increases. In the open labyrinth, our agent is able to reach 100% of possible states (a total of $19 \times 19 = 361$ unique states) in approximately 800 steps, and 80% of possible states ($\approx 290$ states) in approximately 500 steps. These counts also include the $n_{init}$ number of random steps taken preceding training.

Our agent is also able to learn highly interpretable abstract representations in very few environment steps (as shown in Figure 1a) as it explores its state space. In addition, after visiting most unseen states in its environment, our agent tends to uniformly explore its state space due to the nature of our novelty heuristic. A visualisation of this effect is available in Appendix H.

#### 5.1.2 4-room labyrinth

We now consider the 4-room labyrinth environment, a more challenging version of the open labyrinth environment (Figure 1a). As before, our encoder $\hat{e}$ is able to take a high-dimensional input and compress it to a low-dimensional representation. In the case of both labyrinth environments, the representation incorporates knowledge related to the position of the agent in 2-dimensions that we call *primary features*. In the 4-room labyrinth environment, it also has to learn other information

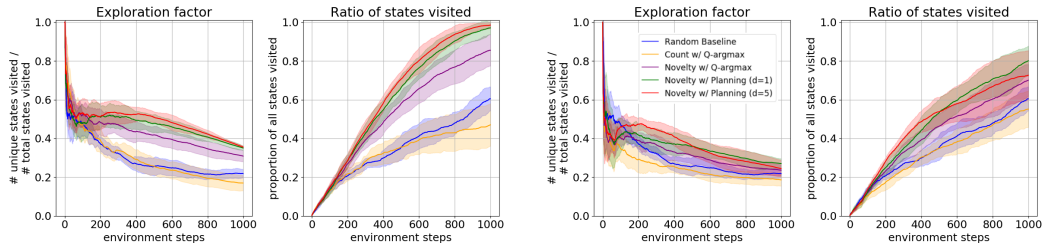

(a) Results for open labyrinth and different variations on policies compared to baselines.

(b) Results for the 4-room labyrinth and different variations on policies compared to baselines.

Figure 2: Labyrinth results for both open labyrinth and 4-room labyrinth over 10 trials, showing mean and standard deviations.

such as agent surroundings (walls, open space) etc., but it does so only via the transition function learned through experience. We call this extraneous but necessary information *secondary features*. As most of these secondary features are encoded only in the dynamics model $\hat{\tau}$, our agent has to experience a transition in order to accurately represent both primary and secondary features.

In this environment specifically, our dynamics model might over-generalize for walls between rooms and can sometimes fail at first to try out transitions in the passageways between rooms. However, because our agent tends to visit uniformly all the states that are reachable within the known rooms, the $\epsilon$-greedy policy of our approach still ensures that the agent explores passageways efficiently even in the cases where it has over-generalized to the surrounding walls.

We run the same experiments on the 4-room labyrinth domain as we do on the open labyrinth and report results in Figure 2b. In both cases, our method outperforms the two baselines in this domain (random and count-based).

## 5.2 Control and sub-goal exploration

In order to test the efficacy of our method beyond fixed mazes, we conduct experiments on the control-based environment Acrobot (Brockman et al., 2016) and a multi-step maze environment. Our method (with planning depth $d = 5$) is compared to strong exploration baselines with different archetypes:

1. Prediction error incentivized exploration (Stadie et al., 2015)

2. Hash count-based exploration (Tang et al., 2016)

3. Random Network Distillation (Osband et al., 2017)

4. Bootstrap DQN (BDQN, Osband et al. (2016))

In order to maintain consistency in our results, we use the same deep learning architectures throughout. Since we experiment in the deterministic setting, we exclude baselines that require some form of stochasticity or density estimation as baselines (for example, Shyam et al. (2018) and Osband et al. (2017)). A specificity of our approach is that we run multiple training iterations in between each environment step for all experiments, which allows the agent to use orders of magnitude less samples as compared to most model-free RL algorithms (all within the same episode).

### 5.2.1 Acrobot

We now test our approach on Acrobot (Brockman et al., 2016), which has a continuous state space unlike the labyrinth environment. We specifically choose this control task because the nature of this environment makes exploration inherently difficult. The agent only has control of the actuator for the inner joint and has to transfer enough energy into the second joint in order to swing it to its goal state. We modify this environment so that each episode is at most 3000 environment steps. While this environment does admit an extrinsic reward, we ignore these rewards entirely. To measure the performance of our exploration approach, we measure the average number of steps per episode that the agent takes to move its second joint above a given line as per Figure 3a.

To demonstrate the ability of our method to learn a low dimensional abstract representation from pixel inputs, we use 4 consecutive pixel frames as input instead of the 6-dimensional full state

| | Acrobot | | | Multi-step Maze | | | Norm. & Combined | | |
|---|---|---|---|---|---|---|---|---|---|
| Reward | Avg | StdErr | p-value | Avg | StdErr | p-value | Avg | StdErr | p-value |
| Random | 1713.3 | 316.25 | 0.0077 | 1863.3 | 308.35 | 0.0025 | 3.26 | 0.41 | $3.1e^{-5}$ |
| Pred | 932.8 | 141.54 | 0.050 | 1018.0 | 79.31 | $4.0e^{-4}$ | 1.78 | 0.15 | $1.3e^{-4}$ |
| Count | 1007.0 | 174.81 | 0.050 | 658.8 | 71.73 | 0.23 | 1.50 | 0.18 | 0.019 |
| RND | 953.8 | 85.98 | 0.0042 | 938.4 | 135.88 | 0.024 | 1.72 | 0.15 | $3.5e^{-4}$ |
| BDQN | 592.5 | 43.65 | 0.85 | 1669.1 | 291.26 | 0.0046 | 2.11 | 0.37 | 0.0099 |
| Novelty | **576.0** | 66.13 | - | **524.6** | 73.24 | - | **1.00** | 0.090 | - |

Table 1: Number of environment steps necessary to reach the goal state in the Acrobot and the multi-step maze environments (lower is better). Results are averaged over 5 trials for both experiments. Best results are in bold. We provide p-values indicative of the null hypothesis $H_0 : \Delta\mu = \mu_1 - \mu_2 = 0$, calculated using Welch's t-test, all as per (Colas et al., 2019). In this case, we do a pair-wise comparison between the central tendencies of our algorithm (Novelty) and our baselines. Normalized and combined results are also shown - results here are first normalized with respect to the average number of steps taken for our algorithm and then combined on both environments.

vector. We use a $4$-dimensional abstract representation of our state and results from experiments are shown in Table 1. Our method reaches the goal state more efficiently than the baselines.

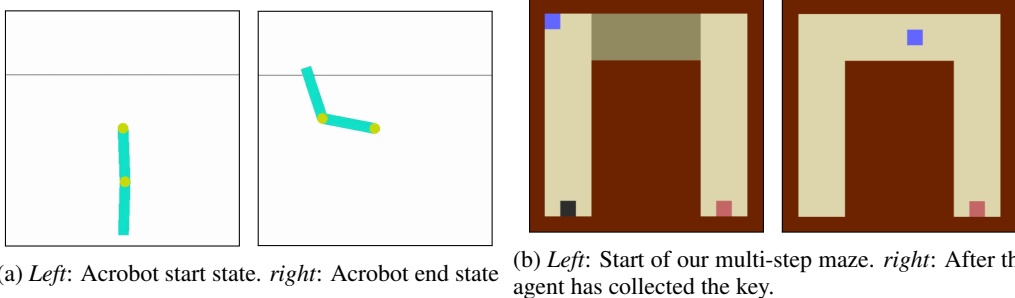

(a) *Left*: Acrobot start state. *right*: Acrobot end state

(b) *Left*: Start of our multi-step maze. *right*: After the agent has collected the key.

Figure 3: Illustrations of the Acrobot and multi-step goal maze environments. *a) Left:* The Acrobot environment in one configuration of its start state. *a) Right:* One configuration of the ending state of the Acrobot environment. The environment finishes when the second arm passes the solid black line. *b) Left:* The passageway to the west portion of the environment is blocked before the key (black) is collected. *b) Right:* The passageway is traversable after collecting the key, and the reward (red) is then available. The environment terminates after collecting the reward.

### 5.2.2   Multi-step goal maze

We also test our method on a more complex maze with the sub-task of picking up a key that opens the door to an area with a reward. We build our environment with the Pycolab game engine (Stepleton, 2017). The environment can be seen in Figure 3b, where the input to our agent is a top-down view of the environment. While this environment does admit an extrinsic reward (1 for picking up the key, 10 for reaching the final state), we ignore these rewards and only focus on intrinsic rewards.

In our experiments, we show that our agent is able to learn an interpretable representation of the environment in a sample-efficient manner. Figure 1c shows an example of learnt representations in this domain after reaching the goal - we observe that positions in the maze correspond to a nearly identical structure in the lower-dimensional representation. Our representation also nicely captures internal state information (whether the key has been picked up) by separating the two sets of states (states when the key has been picked up and states when the key has not been picked up). Similar positions in both sets of states are also mapped closely together in lower-dimensional space (ie. (1, 1, *with key*) is close in $\ell_2$ to (1, 1, *without key*)), suggesting good generalization between similar states.

# 6 Related work

The proposed exploration strategy falls under the category of directed exploration (Thrun, 1992) that makes use of the past interactions with the environment to guide the discovery of new states. This work is inspired by the Novelty Search algorithm (Lehman and Stanley, 2011) that uses a nearest-neighbor scoring approach to gauge novelty in policy space. Our approach leverages this scoring to traverse dynamics space, which we motivate theoretically. Exploration strategies have been investigated with both model-free and model-based approaches. In Bellemare et al. (2016) and Ostrovski et al. (2017), a model-free algorithm provides the notion of novelty through a pseudo-count from an arbitrary density model that provides an estimate of how many times an action has been taken in similar states. Recently, Taiga et al. (2020) do a thorough comparison between bonus-based exploration methods in model-free RL and show that architectural changes may be more important to agent performance (based on extrinsic rewards) as opposed to differing exploration strategies.

Several exploration strategies have also used a model of the environment along with planning. Hester and Stone (2012) employ a two-part strategy to calculate intrinsic rewards, combining model uncertainty (from a random-forest based model) and a novelty reward based on $L_1$ distance in feature space. A strategy investigated in Salge et al. (2014); Mohamed and Rezende (2015); Gregor et al. (2016); Chiappa et al. (2017) is to have the agent choose a sequence of actions by planning that leads to a representation of state as different as possible to the current state. In Pathak et al. (2017); Haber et al. (2018), the agent optimizes both a model of its environment and a separate model that predicts the error/uncertainty of its own model. Burda et al. (2018a) similarly uses an intrinsic reward based on the uncertainty of its dynamics model. In Shyam et al. (2018), forward models of the environment are used to measure novelty derived from disagreement between future states. Still and Precup (2012) take an information theoretic approach to exploration, that chooses a policy which maximizes the predictive power of the agent's own behavior and environment rewards. In Badia et al. (2020), an intrinsic reward from the k-NN over the agent's experience is also employed for exploration. They instead employ a self-supervised inverse dynamics model to learn the embeddings as opposed to our approach. Beyond improved efficiency in exploration, the interpretability of our approach could also lead to human-in-the-loop techniques (Mandel et al., 2017; Abel et al., 2017) for exploration, with the possibility for the agent to better utilize feedback from interpretability of the agent in representation space.

# 7 Discussion

In this paper, we formulate the task of dynamics learning in MBRL through the *Information Bottleneck* principle. We present methods to optimize the IB equation through low-dimensional abstract representations of state. We further develop a novelty score based on these learnt representations that we leverage as an intrinsic reward that enables efficient exploration. By using this novelty score with a combination of model-based and model-free approaches for planning, we show more efficient exploration across multiple environments with our learnt representations and novelty rewards.

As with most methods, our approach also has limitations. One limitation we may have is the scalability of non-parametric methods such as k-NN density estimation since our method scales linearly with the number of environment steps. A possible solution to this problem would be to use some sampling scheme to sample a fixed number of observations for calculation of our novelty heuristic. Another issue that has arisen from using very low-dimensional space to represent state is generalization. In some cases, the model can over-generalize with the consequence that the low-dimensional representation loses information that is crucial for the exploration of the entire state space. An interesting direction for future work would be to find ways of incorporating secondary features such as those mentioned in Section 5.1.2. An interesting possibility would be to use a similar IB method, but using a full history of states as the conditioning variable. Beyond these points, we discuss limitations and potential improvements to this work in Appendix K.

Finally, we show preliminary results of our method on a more complex task - *Montezuma's Revenge* - in Appendix G. With the theory and methods developed in this paper, we hope to see future work done on larger tasks with more complex environment dynamics.

## Broader Impact

Algorithms for exploring an environment are a central piece of learning efficient policies for unknown sequential decision-making tasks. In this section, we discuss the wider impacts of our research both in the Machine Learning (ML) field and beyond.

We first consider the benefits and risks of our method on ML applications. Efficient exploration in unknown environments has the possibility to improve methods for tasks that require accurate knowledge of its environment. By exploring states that are more novel, agents have a more robust dataset. For control tasks, our method improves the sample efficiency of its learning by finding more novel states in terms of dynamics for use in training. Our learnt low-dimensional representation also helps the interpretability of our decision making agents (as seen in Figure 1). More interpretable agents have potential benefits for many areas of ML, including allowing human understandability and intervention in human-in-the-loop approaches.

With such applications in mind, we consider societal impacts of our method, along with potential future work that could be done to improve these societal impacts. One specific instance of how efficient exploration and environment modeling might help is in disaster relief settings. With the incipience of robotic systems for disaster area exploration, autonomous agents need to efficiently explore their unknown surroundings. Further research into scaling these MBRL approaches could allow for these robotic agents to find points of interest (survivors, etc.) efficiently.

One potential risk of our application is safe exploration. Our method finds and learns from states that are novel in terms of its dynamics. Without safety mechanisms, our agent could view potentially harmful scenarios as novel due to the rarity of such a situation. For example, a car crash might be seen as a highly novel state. To mitigate this safety concern we look to literature on Safety in RL (García and Fernández, 2015). In particular, developing a risk metric based on the interpretability of our approach may be an area of research worth developing.

## Acknowledgements

We would like to thank Emmanuel Bengio for the helpful discussions and feedback on early drafts of this work. We would also like to thank all the reviewers for their constructive and helpful comments.

## Footnotes

[1]`https://github.com/taodav/nsrs`

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
