[Supplementary Material · camera_ready_nsrs_neurips2020_14-end.pdf]

## A  Using $\omega$ to gauge model accuracy

The hyperparameter $\omega$ can be used to estimate the accuracy of our transition loss, hence of our novelty estimates. In order to gauge when our representation is accurate enough to use our novelty heuristic, we use a function of this hyperparameter and transition loss to set a cutoff point for accuracy to know when to take the next environment step. If $\omega$ is the minimum distance between successive states, then when $L_\tau \leq \left(\frac{\omega}{\delta}\right)^2$, the transitions are on average within a ball of radius $\frac{\omega}{\delta}$ of the target state. Here $\delta > 1$ is a hyperparameter that we call the slack ratio. Before taking a new step in the environment, we keep training all the parameters with all these losses until this threshold is reached and our novelty heuristic becomes useful. The abstract representation dimensionality is also another hyperparameter that requires tuning, depending on the complexity of the environment. Details on the slack ratios, abstract representation dimensionality and other hyperparameters are given in Appendix J.

## B  Discussion on the distance between successive encoded states

As for our soft constraints on representation magnitude, we use a local constraint instead of a global constraint on magnitude such that it is more suited for our novelty metric. If we are to calculate some form of intrinsic reward based on distance between neighboring states, then this distance needs to be non-zero and ideally consistent as the number of unique states in our history increases. In the global constraint case, if the intrinsic rewards decrease with an increase in number of states in the agent's history, then the agent will fail to be motivated to explore further. Even though the entropy maximization losses ensures the maximization of distances between random states, if we have $|\mathcal{B}|$ number of states in the history of the agent, then a global constraint on representation magnitude might lead to

$$\lim_{|\mathcal{B}|\to\infty} \mathbb{E}_{(s,s')\sim(\mathcal{B},\mathcal{B})}[\|s - s'\|_2] = 0. \tag{9}$$

We also test the necessity of this loss in Appendix F.1 and see that without this loss, we incur a high variance in exploration performance.

## C  Motivation for $\ell_2$ distance

We consider the consecutive distance loss $L_{csc}$. Minimization of this loss ensures that the distance between two consecutive states is $\leq \omega$. This along with our alignment and uniformity losses, $L_{\hat{\tau}}$ and $L_{d1}$ ensures that temporally close states are close in representational space and states are uniformly spread throughout this space. This implies that the minima of $L_{csc}$ between two consecutive states $s$ and $s'$ will occur when:

$$\begin{aligned}
L^*_{consec} &= \min_{\theta_{\hat{e}}} L_{csc}(\theta_{\hat{e}}) \\
&= \min_{\theta_{\hat{e}}} max(\|\hat{e}(s;\theta_e) - \hat{e}(s';\theta_e)\|_2 - \omega, 0) \\
&= \min_{\theta_{\hat{e}}} \left[\|\hat{e}(s;\theta_e) - \hat{e}(s';\theta_e)\|_2 - \omega\right]
\end{aligned}$$

The minimum of this loss is obtained when the $\ell_2$ distance between $s$ and $s'$ is $\omega$. When this loss is minimized, $\ell_2$ distance is well-defined in our representation space, which implies that our novelty heuristic will also be well-defined. These losses shape abstract state space so that $\ell_2$ norm as a distance measure encodes a notion of closeness in state space that we leverage in our novelty heuristic.

## D  Novelty heuristic as an inverse recoding probability score

Consider $P(X_{n+1} = x \mid X_{1:n} = x_{1:n})$, the *recoding probability* of state $x$ at timestep $n + 1$. We try to show that our novelty heuristic of a state is inversely proportional to its recoding probability, or that:

$$\rho_{\mathcal{X}}(x) = \frac{c}{P(X_{n+1} = x \mid X_{1:n} = x_{1:n})}$$

where $c$ is some constant. If we were to try and estimate our recoding probability first using a non-parametric method then using its inverse, we might consider the K-nearest neighbor approach

(Loftsgaarden and Quesenberry, 1965):

$$P(X_{n+1} = x \mid X_{1:n} = x_{1:n}) \approx \frac{k}{nV_{x,x_k}}$$

where $k < n$ is an integer and $V_{x,x_k}$ is the volume around $x$ of radius $d(x, x_k)$, where $x_k \in \mathcal{X}$ is the $k$th nearest neighbor to $x$. The issue with this approach is that our score is only dependent on it's $k$th nearest neighbor (as this score only depends on $V_{x,x_k}$), and doesn't take into account the other $k-1$ nearest neighbors. In practice, we instead try to find something proportional to the inverse directly: we average each of the 1-nearest neighbor density inverses over the k-nearest neighbors:

$$\begin{aligned} \rho_{\mathcal{X}}(x) &= \frac{c}{P(X_{n+1} = x \mid X_{1:n} = x_{1:n})} \\ &\approx nV_{x,x_1} \\ &\approx \frac{n}{k} \sum_{i=1}^{k} V_{x,x_i} \end{aligned}$$

Since we're only worried about proportionality, we can remove the constant $n$ and replace our volume of radius between two points with a distance metric $d$:

$$\rho_{\mathcal{X}}(x) \propto \hat{\rho}_{\mathcal{X}}(x) = \frac{1}{k} \sum_{i=1}^{k} d(x, x_i).$$

Which is our novelty heuristic.

## E  Limiting behavior for novelty heuristic

*Proof (Theorem 1).* Let $n_s$ be the visitation count for a state $s \in S$. We assume that our agent's policy will tend towards states with higher rewards. Given the episodic nature of MDPs, we have that over multiple episodes all states communicate. Since our state space is finite, we have that

$$\lim_{n \to \infty} n_s = \infty, \; \forall s \in S.$$

which means that $\exists n$ such that $k < n_s$ as $n \to \infty$, and implies that the $k$ nearest neighbors of $s$ are indiscernible from $s$. Since $f$ is a deterministic function, $x_i = x$ for all $i$. We also assume that our agent's policy will tend towards states with higher rewards. As $x$ and $x_i$ are indiscernible and $dist$ is a properly defined metric, $dist(x, x_i) = 0$ for all $i$, and we have

$$\lim_{n \to \infty} \hat{\rho}(x) = \frac{1}{k} \sum_{i=1}^{k} dist(x, x_i) \tag{10}$$

$$= 0. \tag{11}$$

$\square$

## F  Ablation study

Here we perform ablation studies to test the affects of our losses and hyperparameters.

### F.1  Consecutive distance loss

To further justify our use of the $L_{csc}$ loss, we observe the results of running the same trials in the simple maze environment (with no walls) with no $L_{csc}$ loss in Figure 4. As we increase the number of forward prediction steps the exploration is less effective and variance of our results increases. Without the relative distance constraints of our representation, we observe an increase of forward prediction errors, which is the likely cause of the decrease in performance. These forward prediction errors are further compounded as we increase the number of forward prediction steps (as can be seen when comparing the standard error between $d = 0, 1, 5$).

Figure 4: Simple maze (with no walls) experiment with no $L_{csc}$ loss.

| Ablation | Avg ($\mu$) | StdErr | p-value |
|---|---|---|---|
| MF | 758.60 | 169.08 | 0.25 |
| MB | 584.10 | 64.52 | 0.57 |
| Full | 524.60 | 73.24 | - |

Table 2: A further ablation study on the multi-step maze environment. The MF (model-free) ablation does not employ any forward intrinsic reward planning ($d = 0$), while the MB (model-based) ablation only uses forward intrinsic reward planning without using or learning Q-values.

## F.2 Pure model-based/model-free

We test the importance of using a combination of both model-based and model-free components on the multi-step maze environment introduced in Section 5.2.2. We train with the same hyperparameters but in the model free ($d = 0$) and model-based ($d = 5$, no Q-value tails) settings. We show results in Table 2.

## G Montezuma's Revenge visualizations

We show preliminary results for learning abstract representations for a more complex task, *Montezuma's Revenge*. Comparing the two figures above, we observe how temporally closer states are closer together in lower-dimensional learnt representational space as compared to pixel space. Transitions are not shown for raw observations.

(a) 5 dimensional abstract representations visualized with t-SNE.

(b) Original observations (each shaped 4 x 64 x 64) visualized with t-SNE.

Figure 5: a) Visualization for 100 observations (4 frames per observation) of Montezuma's Revenge game play. Representation learnt was $n_\mathcal{X} = 5$ and visualized with t-SNE (van der Maaten and Hinton, 2008) in 2 dimensions. Labels on top-left of game frames correspond to labels of states in lower-dimensional space. Transitions are shown by shaded lines. b) Original resized game frames visualized using t-SNE with the same parameters.

# H  Labyrinth state count visualization

Figure 6: An example of the state counts of our agent in the open labyrinth with $d = 5$ step planning. Titles of each subplot denotes the number of steps taken. The brightness of the points are proportional to the state visitation count. The bright spots that begins after 200 counts is the agent requiring a few trials for learning the dynamics of labyrinth walls.

# I Gridworld visualizations

Figure 7: *Left*: Open labyrinth - A $21 \times 21$ empty labyrinth environment. *Right*: 4-room labyrinth - A $21 \times 21$ 4-room labyrinth environment inspired by the 4-room domain in Sutton et al. (1999).

# J Experimental setup and hyperparameters

For all of our experiments, we use a batch size of 64 and take 64 random steps transitions before beginning training. We also use the same discount factor for all experiments ($\gamma = 0.8$) and the same freeze interval for target parameters 1000. The reason behind our low discount factor is due to the high density of non-stationary intrinsic rewards in our state. We also use a replay buffer size corresponding to the maximum number of steps in each environment for all experiments. For all model-based abstract representation training, the following hyperparameters were all kept constant: minimum distance between consecutive states $\omega = 0.5$, slack ratio $\delta = 12$ and transition model dropout of $0.1$. For all experiments run with our novelty metric, we use $k = 5$ for our k-NN calculations. For all experiments that allows for forward planning and not explicitly mention depth $d$, we set planning depth $d = 5$. For abstract representation size ($n_{\mathcal{X}}$), we use a dimensionality of 2 for both labyrinth exploration tasks, a dimensionality of 4 for Acrobot, and finally a dimensionality of 3 for the multi-step maze.

## J.1 Neural Network Architectures

For reference, 'Dense' implies a full-connected layer. 'Conv2D' refers to a 2D convolutional layer with stride 1. 'MaxPooling2D' refers to a max pooling operation. All networks were trained with the RMSProp optimizer. Throughout all experiments, we use the following neural network architectures:

### J.1.1 Encoder

For all our non-control task inputs, we flatten our input and use the following feed-forward neural network architecture for $\hat{e}$:

- Dense(200, activation='tanh')
- Dense(100, activation='tanh')
- Dense(50, activation='tanh')
- Dense(10, activation='tanh')
- Dense(abstract representation dimension).

For our control task, we use a convolution-based encoder:

- Conv2D(channels=8, kernel=(3,3), activation='tanh')
- Conv2D(channels=16, kernel=(3,3), activation='tanh')
- MaxPool2D(pool size=(4,4))
- Conv2D(channels=32, kernel=(3,3), activation='tanh')
- MaxPool2D(pool size=(3,3))
- Dense(abstract state representation dimension).

### J.1.2 Transition model

The input to our transition model is a concatenation of an abstract representation and an action. We use the following architecture

- Dense(10, activation='tanh', dropout=0.1)
- Dense(30, activation='tanh', dropout=0.1)
- Dense(30, activation='tanh', dropout=0.1)
- Dense(10, activation='tanh', dropout=0.1)
- Dense(abstract representation dimension)

and add the output of this to the input abstract representation.

### J.1.3 Reward and discount factor models

For both reward and discount factor estimators, we use the following architecture:

- Dense(10, activation='tanh')
- Dense(50, activation='tanh')
- Dense(20, activation='tanh')
- Dense(1).

### J.1.4 Q function approximator

We use two different architecture based on the type of input. If we use the concatenation of abstract representation and action, we use the following architecture:

- Dense(20, activation='relu')
- Dense(50, activation='relu')
- Dense(20, activation='relu')
- Dense($n_{actions}$)

For the pixel frame inputs for our control environments, we use:

- Conv2D(channels=8, kernel=(3,3), activation='tanh')
- Conv2D(channels=16, kernel=(3,3), activation='tanh')
- MaxPool2D(pool size=(4,4))
- Conv2D(channels=32, kernel=(3,3), activation='tanh')
- MaxPool2D(pool size=(3,3))
- Dense($n_{actions}$).

Finally, for our (purely model free) gridworld environments we use:

- Dense(500, activation='tanh')
- Dense(200, activation='tanh')
- Dense(50, activation='tanh')
- Dense(10, activation='tanh')
- Dense($n_{actions}$)

As for our Bootstrap DQN implementation, we use the same architecture as above, except we replace the final Dense layer with 10 separate heads (each a Dense layer with $n_{actions}$ nodes).

### J.2 Labyrinth environments

Both environments used the same hyperparameters except for two: we add an $\epsilon$-greedy ($\epsilon = 0.2$) policy for the 4-room maze, and increased $n_{freq}$ from 1 to 3 in the 4-room case due to unnecessary over-training. We have the following hyperparameters for our two labyrinth environments:

- $n_{iters} = 30000$
- $\alpha = 0.00025$

### J.3 Control environment

In our Acrobot environment, the input to our agent is 4 stacked consecutive pixel frames, where we reduce each frame down to a $32 \times 32$ pixel frame. Our abstract representation dimension is 4. We use a learning rate of $\alpha = 0.00025$ for all experiments. We train for $n_{iters} = 50000$ for all experiments with the exception of RND and transition loss - this discrepancy is due to time constraints for the latter two experiments which used $n_{iters} = 3000$, as both these experiments used prohibitively more time to run due to the increased number of steps used to reach the goal state of the environment.

### J.4 Multi-step maze environment

In our multistep maze environment, the input to our agent is a single $15 \times 15$ frame of an overview of the environment. Our abstract representation dimension is 3. We use an $\epsilon$-greedy ($\epsilon = 0.1$) policy for this environment. We use $\alpha = 0.00025, n_{iters} = 30000$ for our model-free algorithms and $\alpha = 0.000025, n_{iters} = 50000$ for experiments that include a model-based component. Each episode is at most $4000$ environment steps.

## K   Potential improvements and future work

### K.1   Incorporating agent history for better generalization

As mentioned in Section 5.1.2, generalization across states while only conditioning on *primary features* ($X, A$ in our case) restricts the generalization ability of our agent. An interesting potential for future work would be to somehow incorporate this information into the learnt representation (potentially by using the same IB method, but using a full history of states as the conditioning variable).

### K.2   Increasing efficiency of learning the abstract wtate representations

Currently, learning our low-dimensional state representation takes many iterations per timestep, and is also sensitive to hyperparameter tuning. Our method requires an accurate state representation and dynamics model according to our losses for our method to be effective - the sample-efficiency from our model-learning methods comes at a cost of more time and compute. This is due to the careful balance our model needs to maintain between its losses for good representations. Another interesting potential for future work would be to find ways to incorporate our model-learning losses using less time and computation.

### K.3   Extension to stochastic environments

One avenue of future work would be to extend this work for stochastic environments. While there has been recent work on using expectation models for planning (Wan et al., 2020) that we could use to extend our algorithm, this still comes with its own limitations.

### K.4   Empirical validation for representation size

Another avenue of investigation is to find a more principled approach to finding the right representation size for a given environment. While we currently simply pick the lowest representation size from prior knowledge about an environment, it may be worthwhile to somehow allow the algorithm to decide this.