[Reviews · NeurIPS 2020]

Review 1

Summary and Contributions: This paper proposes a novel exploration algorithm that computes the novelty of a state in a learned representation space and uses it as a reward in a mix of model-free and model-based approach. The main novelty introduced in this paper is the method to learn a representation from which the novelty measure is meaningful.

Strengths: The claims are sound and well justified. The paper is well organized and well written. The empirical evaluation studies several environments. Efficient exploration is a hot topic in RL. The code and all necessary details are provided.

Weaknesses: he main weaknesses of the paper are: * the lack of comparisons to other model-based exploration approaches to disentangle the impact of the model-based aspect vs the novelty measure on performance. * the lack of empirical comparisons and discussion of ICM (Pathak et al., 2017) and RND (Burda, 2018). * the lack of statistical tests to support empirical comparisons. * the lack of an ablative study with respect to the 6 different losses used by the proposed approach.

Correctness: Claims and method are technically sound and well detailed.

Clarity: The paper is well written and well organized.

Relation to Prior Work: The related work section is well developed. However, I would have appreciated a deeper comparison with ICM (Pathak et al., 2017). ICM also learns a representation via a dynamics prediction task (inverse in their case), and uses the prediction-error in a forward prediction task as a measure of novelty. As the prediction task is based on features learned in the inverse dynamics prediction task, the novelty measure is not computed from observations directly, and thus does not have the pitfall described in L20. Moreover, ICM is shown to work on tasks that are more complex than the one presented in this paper and thus appears like a serious contender. Similar comments could be made for RND from Burda et al. 2018.

Reproducibility: Yes

Additional Feedback: The method seems to be restricted to deterministic environments. Could we add a bit of discussion why it would be the case and how we could imagine to extend the approach to deal with stochastic environments (maybe in the supplementary material)? In most approaches, the discount factor is an exponential function of the distance in time, why did the authors choose to make it a function of state and action, and why should we learn it? Having the environment return the discount factor is not really common. The choice of the learned representation size seems to contain some domain knowledge. Can the authors discuss its importance, how it should be selected, and whether the approach is sensitive to this choice? The explanations of the model-based / model-free approach would be better in the main document. It is part of the algorithm, and probably an important part of its success. As such, the algorithms whose performance are presented in Figure 2 are not clearly defined in the main paper. The optimization process includes 6 losses. From eq. 6, it seems that most of them are used to update the encoder representation. I guess that optimizing Lr requires extrinsic rewards but this is not clearly stated. It’s also unclear how all these losses are balanced w.r.t. each other, or how much they contribute to the overall result. With so many moving parts, I would prefer to see an ablative study showing the relative importance of each loss (although Ld1, Lcsc make sense from a theoretical point of view, do we need them empirically?). As such, we cannot know whether all losses contribute equally to the updates of the encoder, or whether some dominate on others. This depends on how these losses scales w.r.t. each other. The experimental section does not include any statistical test. As was shown in [1], using a low number of seeds (e.g. the 5 seeds used here) and comparing the central tendency (e.g. mean) can lead to false conclusions. Claims about algorithms performing better than others should be supported by statistical evidence (statistical tests). See [2] for guidelines. In this paper, these statistical tests would probably pass, which would reinforce the empirical claims. The results only compare the proposed approach with model-free exploration algorithms. This is a problem, as it is not clear whether the performance boost comes from the modeling of the novelty or from the model-based component of the proposed algorithm. To disentangle the two, we need a comparison with a model-based exploration approach. One could use an existing one (e.g. [3, 4]) or modify a model-free one (e.g. ICM, replacing the model-free RL part of ICM by an MPC). An alternative could be to show the performance of the proposed approach in a pure model-free setting (e.g. replacing MPC by vanilla DQN). I would like a deeper discussion about what prevents this algorithm to scale to more difficult tasks (e.g. Montezuma’s revenge) and how these difficulties could be overcome in the future. **Minor comments**: * The opening quotes are not typed correctly. The opening double quote should be twice the ` character. * L610: model-base → model-based * From Eq 3, it seems the dynamics function tau computes the delta x’ - x instead of x’. This should be said explicitly. * H is used for entropy and for the history. Maybe use \mathcal{B} and call it replay buffer? Overall, I’ll give a 6 to this paper. I would be willing to increase that score if the paper was revised with statistical tests, comparisons to a model-based exploration approach (or removing the model-based component of the proposed approach) and an improved discussion of the comparison with ICM and the relative importances of the 6 losses in Eq 6. **References**: [1] Deep Reinforcement Learning that Matters (Henderson et al., 2018) [2] A Hitchhiker's Guide to Statistical Comparisons of RL Algorithms (Colas et al., 2019) [3] Deep RL in a Handful of Trials (Chua et al., 2018) [4] Model-based active exploration. (Shyam, 2019). **Update post rebuttal** In the above review, I listed 4 weaknesses and explicitly said I would revise my score if these were addressed: 1. Lack of statistical tests. The authors doubled the number of seeds doubled and conducted statistical tests whose results supported the claims. 2. Lack of empirical comparisons to RND and ICM. These were discussed a bit more in the authors' response. 3. Lack of comparisons to model-based exploration approaches. The authors added an ablation of the model-based component. 4. Lack of discussion / ablation for the loss with 6 components. The authors added a discussion of the discount factor learning loss. They also provided clarifications about other minor comments of mine. Overall, I think learning embeddings in which Novelty Search can be performed is an interesting problem to tackle. It does not work perfectly on Montezuma's Revenge yet, but I believe it should be published, so that others see it and scale it to harder problems. I update my score to 7.


Review 2

Summary and Contributions: This paper proposes a model based exploration method based on an encoding of the environment. The method creates a low dimensional representation of the space and then rewards the agent for seeking novelty based states that are farther apart in dynamics. Results are tested empirically in multiple grid-type domains to evaluate stage coverage, as well as two control domains to evaluate the improvement of the search for novelty on the agent’s ability to perform well on control tasks. In both the grid domains and the control domains the agent performed better than the baselines it was compared against.

Strengths: The paper is well written and explains well each of the components of the proposed algorithm very well. The theory (and it’s sub-concepts like Information Bottlenecks) were explained clearly. The environments they chose for empirical evaluation were good demonstrations of the strengths of their method. The evaluation of the results themselves was clear.

Weaknesses: The loss function (eq 6) is extremely long and it is not clear from the experiments if different parts of the loss were more important that others. It would have been nice to have a discussion about this and an ablation study of the different components. The empirical section chose domains that were clear in their ability to show the effects of the method. However only having 5 runs in a small domain lead to the results not being significantly different than each other. It was difficult to know the difference in the methods based on the small number of runs and overlapping error bars. While the paper was genuinely clear some of the components of the method were not clearly motivated. For instance - why parameterize and learn the discount function? It was not clear why this choice was made and the effect it was going to have on the method.

Correctness: Given how small the domains were it would have been nice to have more than 5 runs of each algorithm for comparison. As referenced in Deep Reinforcement Learning that Matters (Henderson, 2019) seeds can play a large role in the performance of the DRL algorithms. In smaller domains more runs ideally will be performed to help distinguish the results of each method from each other. The results in this case are fairly close to each other, with overlapping error bars. It is difficult to draw conclusions from the empirical results based on this and with so few runs.

Clarity: Generally the paper is clearly written, telling a clear story about the method and its results. As mentioned above, some of the pieces of the method could have been more thoroughly motivated.

Relation to Prior Work: Overall the paper does a good job of citing previous work and placing their method in the context of previous work. Two things that would be useful: Referencing and discussing Barto’s work “Novelty of Surprise?”. This work distinguishes the two. Given how much exploration/curiosity work is focused on surprise, with this work focused on novelty, it would be nice to place this in the context that Barto provides. This seems to be building on work done by Still and Precup (An information-theoretic approach to curiosity-driven reinforcement learning). Is this an extension with empirical results of that work? Still’s other work on the topic is cited, but it is unclear how strong the tie between the two works is.

Reproducibility: Yes

Additional Feedback: Some things that would revise my score: More runs and clarity on the statistical significance of the empirical results. Clarity on how it ties into previous work in exploration with Information Bottlenecks (i.e. Still). Clarity on why the different components were chosen and their effect on the results.


Review 3

Summary and Contributions: This paper presents a method for compressing a large state space into a small one that is well-suited for novelty-based exploration. The state space is constructed by maximizing compression while retaining an accurate dynamics model, and additionally enforcing a single-step distance between successive states. This is evaluated qualitatively on some maze domains and then in terms of return on two smaller domains.

Strengths: The approach is relatively simple and clean, and works surprisingly well. This is an extremely important problem where existing approaches are typically a little ad-hoc and problem motivated (which is counter to the point). I think the qualitative demonstrations are very compelling.

Weaknesses: I have two main complaints. One (which can be resolved in rebuttal) is: what exactly is the state space for the environments in Figure 1? I think it's an image, but I don't see where that is explained. The second is that the quantitative experiments on RL tasks are limited to a very small maze and Acrobot, which borders on fatally insufficient.

Correctness: Yes.

Clarity: Yes, though it has a couple of annoying things. There's a consistent casual sloppiness about the writing throughout. For example, there's a lot of "s.t." and "r.v."s dropped, which is not appropriate in formal writing. The dashes between sentence fragments should be m-dashes. There's citation weirdnesses - why mention and cite POMDPs when your method is not applicable to them and has not been tried? Is it really necessary to cite 9 different papers about the idea that a state space might be high-dimensional? Quotation marks are backwards in places. Titles and subsection headings have inconsistent capitalization. All of this leaves a very poor impression. Finally, I think a lot of the interesting stuff is shoved into the Appendix - the algorithm, the planner, and the Monty results should all be in the main text. I know there's a shortage of space, but there are some obvious ways to reduce that (a lot of equations don't need their own line, there could be way less subsectioning, etc.)

Relation to Prior Work: Yes

Reproducibility: Yes

Additional Feedback:


Review 4

Summary and Contributions: This work presents a novel approach for learning low-dimensional encodings of states motivated by the information bottleneck principle. These encodings are leveraged simultaneously for generating novelty-based intrinsic rewards, as well as planning. The method is assessed in terms of state coverage and performance on several toy environments and shown to exhibit consistent improvement over the baselines.

Strengths: The visualizations of the abstract state representations learned in selected gridworld domains, demonstrate a promising ability to capture global geometric structure and environment dynamics The experiments provide convincing evidence that the proposed novelty search and planning improves state coverage and performance in simple cases.

Weaknesses: I think the general method is interesting, but the paper would greatly benefit from experiments on some harder tasks and comparison to stronger baselines. In particular: - The authors include visualizations on Montezuma’s Revenge but do not report performance. Including this would provide more convincing evidence that the proposed method can generalize to more complex domains. - It would be interesting to evaluate this method some in the presence of noisy observations or predictable but uncontrollable dynamics (e.g. a clock in the observation) - I would expect both Random Network Distillation and ICM (Curiosity-driven Exploration by Self-supervised Prediction) to provide stronger baselines for intrinsic motivation than the prediction error or hash-count methods reported in the paper.

Correctness: No errors in claims or methodology were apparent.

Clarity: The overall method and intuition are well-described. However, it’s ambiguous precisely what history of states is used for the KNN computation (e.g. is it episodic?, is there some fixed buffer size?, etc.), and I think some description of this is needed.

Relation to Prior Work: KNN-based intrinsic rewards have previously been explored in works such as Never Give Up. Some discussion of the similarities and difference to this or related works would provide useful context. Part of the loss used for representation learning is closely related to the alignment and uniformity losses in “Understanding Contrastive Representation Learning through Alignment and Uniformity on the Hypersphere", which the authors point out.

Reproducibility: Yes

Additional Feedback: Besides recommendations given in previous section, it could be useful to examine how different choices for the abstract state representations affect planning. (e.g. what are the tradeoffs in terms of the quality of the novelty signal and planning as the encoding size is varied?) # Post-rebuttal The addition of statistical tests, RND baseline, and ablations are sufficient to alleviate my primary concerns. While I would like to see some discussion of scaling issues in the revision, I believe it is worthwhile to publish this work with the proposed revisions, and will adjust my score to a 7 accordingly.

[Author Response · NeurIPS 2020]

We would like to thank all reviewers for their thoughtful comments that have helped improve the paper. We have implemented most of the suggestions and we answer questions below.

Our main additions are as follows. (i) Doubling the number of seeds for experiments (to 10 seeds), and including statistical tests that show the significance of the results. (ii) Providing additional ablations (pure model-free, pure model-based) (as shown in Table B) (iii) Adding a new baseline RND (Burda et al., 2018) in Table A.

| Reward | Avg ($\mu$) | StdErr | p-value |
|---|---|---|---|
| Random | 3.063 | 0.379 | 0.000040 |
| Pred. error | 2.372 | 0.359 | 0.0016 |
| RND | 2.203 | 0.391 | 0.0081 |
| BDQN | 1.859 | 0.264 | 0.0064 |
| Hash-count | 1.304 | 0.209 | 0.20 |
| Novelty | **1.000** | 0.0899 | - |

Table A: Combined results table over both acrobot and multi-step maze environments over 10 random seeds, normalized to the mean number of steps in our Novelty approach for each environment. We provide p-values indicative of the null hypothesis $H_0 : \Delta\mu = \mu_1 - \mu_2 = 0$, calculated using Welch's t-test, all as per Colas et al. (2019). In this case, we do a pair-wise comparison between the central tendencies of our algorithm (Novelty) and our baselines. Full details including these statistical tests will be included in the final paper.

**(All) Ablation study and contribution of the different elements.** The additional ablations in Table B allows assessing the impact of some components of Eq 6. It also shows that using a combination of using a Q-value in combination with a model in the context of exploration is one key contribution as compared to ICM (Pathak et al., 2017) and RND (Burda et al., 2018). Also important, as compared to these works, we provided visualisations of the abstract representations obtained.

**(R1, R2) Learning the discount factor ($\gamma$) and discussion on the different losses used.** In our experiments, learning the discount factor is only used for ensuring correctness in planning for terminal states (where $\gamma = 0$). The loss associated with learning gamma is unlikely to have a significant impact as it decreases rapidly due to the simplicity of learning to map to a constant $\gamma$ everywhere except for terminal transitions where it has to map to 0. We'll clarify that in the paper. We'll also mention that the loss associated with the reward function has been used previously as an auxiliary task that has the potential to *improve* learning, even in a pure model-free setting (where it is not used during planning) (Jaderberg et al., 2016). In summary, some losses in Equation 6 are easy to optimize and are likely to help learning and/or do not require particular tuning in the learning process. We will add these points to the discussion.

**(R2, R4) Additional references.** We will add in our related work section the relevant works of Still and Precup, Barto's work "Novelty of Surprise?" and "Never Give Up" by Badia et al. "Novelty of Surprise?" explains well the motivation of approaches based on novelty, and how they are related to "surprise-based" approaches for exploration. Still and Precup's work is related to our motivation for estimating novelty from an abstract representation that has to contain the minimal meaningful information for representing the environment. Finally, "Never Give Up" further supports our use of kNN as a novelty measure.

| Ablation | Avg ($\mu$) | StdErr | p-value |
|---|---|---|---|
| MF | 944.2 | 136.477 | 0.053 |
| MB | 828.3 | 106.914 | 0.12 |
| Full | 591.2 | 81.86 | - |

Table B: A further ablation study on the multi-step maze environment. The MF (model-free) ablation does not employ any forward intrinsic reward planning ($d = 0$), while the MB (model-based) ablation only uses forward intrinsic reward planning without using or learning Q-values.

**(R1, R2, R4) Choice of the abstract representation.** The choice of the abstract representation and possibly the encoder architecture can be important elements. In practice, we observed that as long as the model allows sufficient capacity (e.g. at least 2 hidden neurons in the open grid world), we did not observe any consistent difference in performance. We will clarify this in the discussion and appendix of the paper.

**(R1, R3) Clarifications & pseudo-code** Figure 1 is obtained from environments with high dimensional observations and the visualizations are provided in Appendix J, which we will explicitly mention in the paper. We have also taken into account the other suggestions for improved clarity, such as bringing the algorithm to the main part of the paper.

**(R1, R3) Deterministic environments** The method as implemented is indeed currently limited to deterministic environments (as mentioned in line 43 of Section 2). That limitation could be relaxed with a generative internal model and by taking into account an expected distance for our novelty metric to handle stochastic domains.

**(R1, R4) Montezuma's revenge.** A full application to Montezuma's revenge is complicated due to the fact that many gradient descent steps are performed at each environment step in order to learn the model with sufficient accuracy. Preliminary results show that our approach provides a meaningful abstract representation that would allow efficient exploration in such complex games (see Appendix H), though at the cost of expensive computations.

**Additional changes.** We will also gladly incorporate changes about cleaning references, formalism and wording.

[Meta-Review · NeurIPS 2020]

This paper proposes an novelty-search exploration method based on an encoding of the environment. Their method computes the novelty of a state in a learned representation embedding space and encourages the agent to optimize for this novelty using a combined model-free and model-based approach. Motivated by the information bottleneck principle, the embedding space is learned by maximizing compression while retaining an accurate dynamics model, resulting in compressing the environment into a small state space well-suited for novelty-based exploration. The experiments were also clear and well-motivated, on grid-type domains to evaluate state coverage, and also two control domains to evaluate the improvement of novelty search on the agent's ability to perform control tasks. I particularly enjoyed the learned abstract visualization of the labyrinth env in Figure 1. Reviewers gave the authors good feedback in terms of references, and suggestions to improve experiments and measurements that were partially addressed in the rebuttal. All in all, everyone agrees that this work should be accepted at NeurIPS and will be a fine contribution. The topic is relevant to wide range of NeurIPS audience and also can be presented nicely in a visual presentation.